# Chronic Effects of Imidacloprid on Honey Bee Worker Development—Molecular Pathway Perspectives

**DOI:** 10.3390/ijms222111835

**Published:** 2021-10-31

**Authors:** Yun-Ru Chen, David T. W. Tzeng, En-Cheng Yang

**Affiliations:** 1Department of Entomology, National Taiwan University, Taipei 10617, Taiwan; milkingroom@gmail.com; 2School of Life Sciences, The Chinese University of Hong Kong, Hong Kong 999077, China; allqwdd@gmail.com

**Keywords:** sublethal dosage, imidacloprid, honey bee, bumble bee, molecular effect

## Abstract

Sublethal dosages of imidacloprid cause long-term destructive effects on honey bees at the individual and colony levels. In this review, the molecular effects of sublethal imidacloprid were integrated and reported. Several general effects have been observed among different reports using different approaches. Quantitative PCR approaches revealed that imidacloprid treatments during the adult stage are expressed as changes in immuneresponse, detoxification, and oxidation-reduction response in both workers and queens. In addition, transcriptomic approaches suggested that phototransduction, behavior, and somatic muscle development also were affected. Although worker larvae show a higher tolerance to imidacloprid than adults, molecular evidence reveals its potential impacts. Sublethal imidacloprid treatment during the larval stage causes gene expression changes in larvae, pupae, and adults. Transcriptome profiles suggest that the population and functions of affected differentially expressed genes, DEGs, vary among different worker ages. Furthermore, an early transcriptomic switch from nurse bees to foragers was observed, suggesting that precocious foraging activity may occur. This report comprehensively describes the molecular effects of sublethal dosages of imidacloprid on the honey bee *Apis mellifera*. The corresponding molecular pathways for physiological and neurological responses in imidacloprid-exposed honey bees were validated. Transcriptomic evidence suggests a global and sustained sublethal impact of imidacloprid on honey bee development.

## 1. Introduction

Neonicotinoids, also known as neonics, are a class of neuroactive pesticides derived from nicotine [1,2]. Their systematic and highly water-soluble nature makes them one of the most commonly used pesticides worldwide [3,4]. Systemic pesticides can be absorbed into plants and transported throughout plant tissues [3,4]. Insects and other pests will be affected after consuming neonics-absorbed plants or any other contaminated sources. As a class of neuroactive pesticides, neonicotinoids act on the nicotinic acetylcholine receptor (nAChR), permanently bind to nerve cells, block neurotransmission, and cause nerve overstimulation. Poisoned insects show symptoms of twitching, paralysis, and eventually death [5,6,7]. Neonics can be applied to seed coatings to prevent storage pests or soil applications to control plant-sucking insects such as aphids and scale insects [3]. Although they are one of the most commonly used pesticides, increasing scientific evidence suggests that environmental residue levels of neonicotinoids cause long-term negative impacts on human-maintained honey bee colonies and wild bees around the world [8,9,10,11,12,13,14,15].

For honey bees, foragers are in contact with pesticides during foraging and are the first affected. Those who survive the poisonous pesticides then return to the beehive with neonicotinoid-contaminated nectar and pollen. Nurse bees, larvae, and even queens are consequently threatened by sublethal neonicotinoids through the consumption of pesticide-contaminated food [16,17,18,19,20,21]. Mitchell et al. (2017) examined honey samples worldwide and found that approximately 75% of the honey samples contained at least 1 neonic, 45% of them contained more than 2 neonics, and 10% of the samples contained more than 4 neonics [19]. This report clearly demonstrates that pollinators around the world are under the threat of pesticides, especially neonicotinoids. To protect pollinators, the outdoor usage of 3 neonicotinoids, imidacloprid, thiamethoxam, and clothianidin, has been banned by the European Union [22] to stop the severe, accumulative damages they generate. The effects of imidacloprid, one of the most widely used neonicotinoids, are discussed in this review.

The lethal dose/concentration of imidacloprid in the honey bee *Apis mellifera* has been widely surveyed, and the levels vary among different regions and seasons (Table 1). In Italy, the acute oral toxicity (AOT) LD_50_ value for imidacloprid at 24 h is 118.74 ng/honey bee, and at 48 and 72 h, it is 90.09 and 69.68 ng/bee, respectively [23]. In France, the LD_50_ (oral application) at 24 h is 5 ng/bee [24], while the LD_50_ (oral application) at 48 and 72 h is 57 ± 28 ng/bee and 37 ± 10 ng/bee, respectively [25]. The oral LD_50_ at 72 h ranged from 20 to 81 ng/bee in Germany, the United Kingdom, and the Netherlands [26]. In Egypt, the LC_50_ (oral application) under laboratory conditions is 3 parts per billion (ppb) at 24 h and 0.6 ppb at 48 h, while the LD_50_ (topical application) is 29 and 26 ng/bee at 24 and 48 h, respectively [27]. Compared to adults, honey bee larvae can tolerate higher dosages of imidacloprid at LD_50_ = 4.17 μg and LC_50_ = 138.84 parts per million (ppm) [28]. Different levels of pesticide sensitivity among different regions may be correlated with environmental factors, such as climate conditions [29], processes of breeding [30], as well as individual differences [31,32]. The level of imidacloprid residue in honey bee bread and wax varies among different areas, from 19.7 ppb to 912 ppb [33]. The concentrations of imidacloprid in dead bees vary from 12 to 223 ng/g dead bee [34]. Imidacloprid residue in bee bread/wax may generate lethal effects if the level is high, and the risk from low residue levels is still a concern. Although a few reports suggest that realistic field doses of imidacloprid pose a low risk for honey bees [35,36], scientific evidence has strongly suggested that a residue level of imidacloprid does not cause immediate death but does generate negative impacts on the ecological sustainability of honey bees, as the impact of chronic pesticide exposure can damage the whole colony or population and even pass to the next generation rather than being limited to individual effects [15,19,35].

Contamination with a sublethal dosage of imidacloprid causes similar but different effects on nurse bees and foragers. For nurse bees, exposure to a sublethal dosage/concentration of imidacloprid impairs their olfactory-associated learning ability and reduces their activity and social interaction [42,43,44]. The acini of the hypopharyngeal glands of workers with 20 to 30 ppb imidacloprid treatment are reduced in size to 14.5% and 16.3% for 9- and 14-day-old nurse bees, respectively [45]. For foragers, exposure to sublethal imidacloprid dosages shortens their lifespan, permanently impairs olfactory-associated learning ability, and alters foraging behavior and foraging frequency [39,46,47,48,49,50,51,52,53]. In addition, respiratory rhythm and metabolic rate are affected. The duration of the bursting pattern of abdominal ventilation movement (AVM) shows a 59.4% increase in the interburst interval and a 56.99% decrease in AVM burst duration [45]. Exposure to 5 ppb imidacloprid significantly increases the metabolic rate of foragers at 25 °C but not at 30 and 35 °C [54]. Imidacloprid exposure also weakens the immune system and the ability of honey bees to protect against parasites. Newly emerged workers showed a reduction in hemocyte density, encapsulation response, and antimicrobial activity after exposure to 1 ppb or 10 ppb imidacloprid [55]. A synergetic effect is observed in the combination of microsporidia and imidacloprid. Exposure to a sublethal level of imidacloprid causes the elevation of *Nosema* spore numbers, and honey bees showed a high individual mortality rate and energetic stress in the presence of both agents [56,57]. Neurologically, sublethal concentrations of imidacloprid exposure during the larval stage cause a reduction in synaptic density in the mushroom body, the brain region responsible for olfactory and visual functions, consequently causing impairment of olfaction-related learning behavior [41,58]. These impairments consequently may make workers vulnerable, even reducing worker lifespan and resulting in colony collapse.

## 2. The Effects of Sublethal Dosages of Imidacloprid Exposure during the Adult Stage from a Molecular Perspective

Behavioral, physiological, and neurological changes may be associated with the expression of corresponding genes or related molecular regulatory pathways. To confirm this, several studies used qPCR to examine the expression of target genes to ensure the molecular effects on encoded genes such as detoxification, antioxidant enzyme production, and immune response [59,60,61,62]. Exposure to 5, 20, and 100 ppb imidacloprid for 7 weeks induced the differential expression of GSTD1 [59]. Exposure to 1 μL of 20 ppb (0.02 ppm) imidacloprid through topical treatment changed the mRNA expression level of P450 superfamily, immunity, and development-related genes in workers. More specifically, genes related to development, antioxidant enzyme coding (catalase, superoxide dismutase (SOD), and thioredoxin peroxidase), and immunity (apidaecin and Amel\LRR) were downregulated at 7 d post-treatment [60]. De Smet et al. examined the expression of genes related to immune end-products, vitellogenin, and detoxification enzymes of honey bees raised either in the laboratory (in cages) or in the field after consuming pollen patties containing 5 ppb and 200 ppb imidacloprid. Their results suggest that honey bees show different responses in different environments. Most of the tested immunity- and vitellogenin-related genes were downregulated in cage-raised bees at both treatment concentrations. In contrast, field bees showed different expression patterns, where 5 ppb imidacloprid treatment generated little or no effect and 200 ppb treatment caused an upregulation in the immune response [61]. The expression of vitellogenin-related genes also was upregulated in bees exposed to imidacloprid under field conditions. Most detoxification enzymes were downregulated after 10 d of exposure to imidacloprid in caged honey bees, with a significant downregulation of CYP9Q3. Conversely, most detoxification enzymes were upregulated under field conditions after 20 d of 200 ppb imidacloprid exposure, with significant upregulation of CYT P450 and CYP9Q3. Some genes were upregulated after 10 d but at low levels. Exposure to lower concentrations of imidacloprid under field conditions had little or no effect on the expression levels of the tested detoxification genes [61]. Gregorc et al. examined the expression of 10 honey bee antioxidant genes related to pesticide toxicity from 5 or 10 ppb imidacloprid and found upregulation of 3 and 4 antioxidant-related genes, respectively [62] (Table 2a).

Quantitative PCR can provide accurate and specific expression profiles of target gene sets, but data are limited and restricted to well-understood pathways. The transcriptomic approach is essential and critical to understanding the comprehensive impacts of imidacloprid on honey bees. Wu et al. (2017) used the next-generation sequencing (NGS) approach to profile the transcriptome of nurse bees with imidacloprid exposure [64]. For 8-day-old nurse bees, exposure to 10 ppb imidacloprid for 8 d resulted in 509 differentially expressed genes (DEGs), with 160 up- and 349 downregulated genes. The upregulated genes were related to the functions of ribosomal proteins (including 18 60S ribosomal proteins and 10 40S ribosomal proteins), phototransduction, visual perception, and photoreception. In contrast, the downregulated genes were related to actin binding, the actin cytoskeleton, muscle attachment, and somatic muscle development. Behavioral observations also confirmed impaired climbing ability [64]. For 11-day-old nurse bees, exposure to 20 ppb imidacloprid for 11 d resulted in 1 and 130 upregulated and downregulated genes, respectively. GO analysis suggests that the downregulated genes were related to detoxification response, immunity, and oxidation-reduction [64] (Table 2a).

Although queen bees are not directly in contact with imidacloprid-contaminated sources, they can still be affected through brooding behavior from nurse bees or from wax. Furthermore, imidacloprid can accumulate in royal jelly, while nurse bees consume contaminated pollen [35]. The effect of imidacloprid on queen bees should thus be examined. In a physiological study, exposure to a sublethal dosage of imidacloprid caused a low fecundity rate and delayed egg laying, worker brood care, and production, and reduced the queen survival rate [68,69]. Imidacloprid-treated queens showed less activity and tended to stay immobile, and the effect was dose-dependent [70]. Exposure to 2.5 ppb imidacloprid for 4 continuous d caused a strong reduction in the standard metabolic rate in queen bees [70]. To confirm the molecular effect, Chaimanee et al. (2016) used qPCR to detect the expression of P450 superfamily genes, immunity, and development/antioxidant enzyme coding after 20 ppb imidacloprid treatment of queen bees with topical treatment during the adult stage. Among the 15 target genes, only CYP4G11 (P450 superfamily) and thioredoxin peroxidase were upregulated, while CYP306A1 (P450 superfamily), CYP6AS14 (P450 superfamily), apidaecin (immunity), eater (immunity), Amel/LRR (immunity), VgMC (immunity), and development and antioxidant enzyme-coding genes SOD and hexamerin 70b were downregulated [59] (Table 2b).

Exposure to sublethal imidacloprid during the adult stages of both workers and queens has been investigated and described. Immunity, detoxification, and antioxidant responses were of high interest and thus targeted in most reports. Although conditions varied, the expression trends of a large proportion of related genes were downregulated. Table 2a shows the concentration/dosage of imidacloprid applied, treatment stages, affected gene lists, and gene regulation trends in honey bees.

## 3. The Effects of Sublethal Dosages of Imidacloprid Exposure during the Larval Stage from a Molecular Perspective

Honey bee larvae are more tolerant to higher concentrations/dosages of imidacloprid than adults, and they can survive treatments and develop into adults [28]. Nevertheless, the larvae showed delay in development. The average eclosion date was 1 to 2 d delayed compared to the control, and the developed adults showed a short lifespan and low survival rate [71,72]. Full-grown adults showed disorders in olfactory associative learning behavior and a high failure rate in foraging behavior [41,73], and a precocious forager was developed [73,74]. Compared to a fully developed, normal forager, an early matured forager showed poor flying skills and less foraging efficiency, and the death rate was higher [74,75,76,77,78]. A neurological study revealed fewer synapses in the mushroom body than in healthy bees, suggesting that neurological damage occurs from the intake of sublethal dosages of imidacloprid [58]. Molecular effects thus need to be verified. Using the qPCR approach, Tesovnik et al. (2019) examined honey bee adult immune response-related genes after exposure to 20 ppb imidacloprid during the larval stage for 4 continuous d. They identified the downregulation of immune responses, including Toll, JAK/STAT, IMD, JNK, and antimicrobial peptides, in both white-eyed and brown-eyed pupae, while those of adults were upregulated [67] (Table 2c).

Using a transcriptomic approach, Wu et al. (2017) revealed the effects on adult workers after exposure to sublethal imidacloprid treatment during their larval stage [66]. Exposure to 500 ppb imidacloprid for 4 continuous d during the larval stage induced 578 differentially expressed genes in 5/6-day-old adults. Among them, 329 genes were annotated and classified into 11 functional groups, while the functions of the remaining 249 genes required further research [66]. The 11 groups were detoxification (24 genes, 7 upregulated, 14 downregulated), immunity (5 genes, all downregulated), mitochondria (11 genes, 7 upregulated, 4 downregulated), metabolism (24 genes, 3 upregulated, 21 downregulated), neuron development (23 genes, 4 upregulated, 19 downregulated), sensory processing (18 genes, 4 upregulated, 14 downregulated), signaling pathways (8 genes, 2 upregulated, 6 downregulated), structural proteins (36 genes, 32 upregulated, 4 downregulated), transcription factors (5 genes, 2 upregulated, 3 downregulated), transporters and receptors (46 genes, 10 upregulated, 36 downregulated), and others (129 genes) (Table 2c). Furthermore, the expression level of genes encoding major royal jelly proteins (MRJPs) was significantly downregulated, suggesting that the composition of royal jelly may be affected and that malnutrition may occur for queen and young larvae [66].

To comprehensively evaluate the molecular effects of imidacloprid on worker development, transcriptomes of 9-day-old larvae and 0-, 7-, 14-, and 20-day-old adults were sequenced after exposure to 1 μL of 1 ppb, 10 ppb, or 50 ppb imidacloprid during the larval stage for 4 consecutive d (2-day-old to 5-day-old larvae) [79]. The numbers of DEGs and the developmental queue showed no significant correlation, and a dosage-dependent effect was observed only on 9-day-old larvae and 0-day-old adults. In 7-day-old adults, DEGs were only identified in bees with 10 ppb treatment during the larval stage, and more than 80% of them were upregulated. The number of DEGs decreased as worker development progressed but then peaked in 14-day-old adults. The DEG numbers were 4871, 5863, and 5848 for the 1, 10, and 50 ppb imidacloprid treatments, respectively. Very few DEGs were identified in 20-day-old adults for all treatments. Further comparisons were performed and revealed high transcriptome similarity between 14-day-old imidacloprid-treated bees and 20-day-old controls. The upregulation of forager regulators, transcription factors that regulate the honey bee behavioral state from nurse to forager, was also identified, suggesting that precocious foragers may develop when exposed to sublethal concentrations of imidacloprid during the larval stage. Common effects among workers of different ages were then investigated by comparing the population of DEGs. As very few DEGs were identified from imidacloprid-treated 20-day-old adults, the shared DEGs were analyzed among 9-day-old larvae and 0-, 7-, and 14-day-old adults. A total of 35 genes were found to be constantly differentially expressed. Two of them were related to the nervous system, including neuroactive ligand receptor (GB40975) and nicotinamide riboside kinase (GB53410). Nicotinamide riboside kinase (Nrk1) can protect damaged neurons from degradation, and its expression is especially induced in the presence of nerve damage [80]. The constant upregulation of Nrk1 confirmed damage to the honey bee neural system after imidacloprid treatment during the larval stage. To understand the effect of imidacloprid on workers of different ages, we will focus on the affected GO terms and pathways at different ages in the following section.

## 4. The Affected Molecular Pathways at Different Ages of Honey Bee Workers

The numbers of DEGs varied among different ages and treatments, without any significant trend. To further understand the function of affected genes, GO analysis and KEGG pathway prediction were performed for DEGs identified at various worker ages. For 9-day-old larvae, upregulated DEGs were significantly enriched in biological terms of translation (GO:0006412, 23 genes, 10 ppb), and the KEGG pathways were ribosome (24 genes, 10 ppb, ame03010) and proteasome (6 genes, 10 ppb, ame03050). For downregulated genes, DEGs were significantly enriched in biological terms of homophilic cell adhesion via plasma membrane adhesion molecules (GO:0007156, 5 genes, 10 ppb) and regulation of Rho protein signal transduction (GO:0035023, 5 genes, 10 ppb). The KEGG pathway was enriched in the Hippo signaling pathway—fly (ame04391, 8 genes, 10 ppb) and insulin resistance (ame04931, 7 genes, 10 ppb). For 0-day-old adults, upregulated DEGs were enriched in biological terms of regulation of transcription, DNA-templated (GO:0006355, 16 genes, 50 ppb) and signal transduction (GO:0007165, 7, 11, and 13 genes for 1, 10, and 50 ppb, respectively), while no significant KEGG pathway was identified. For downregulated DEGs, no significant biological term was identified, and the KEGG pathways were enriched in oxidative phosphorylation (ame00190, 14 genes, 50 ppb) and carbon metabolism (ame01200, 17 genes, 50 ppb). For 7-day-old adults, upregulated DEGs were enriched in translation (GO:0006412, 14 genes, 10 ppb) and ribosome (KEGG pathway ame03010, 18 genes, 10 ppb). In contrast, the downregulated DEGs were enriched in terms related to the regulation of transcription, DNA-templated (GO:0006355, 5 genes, 10 ppb). For 14-day-old adults, more than 4000 DEGs were identified from all treatments; thus, more significant functions/pathways were identified. For upregulated DEGs, biological terms were enriched in transcription, DNA-templated (GO:0006351, 38 genes, 10 ppb) and translation (GO:0006412, 40 genes, 50 ppb), and the KEGG pathways were ribosome (ame03010, 48 genes, 10 ppb), FoxO signaling pathway (ame04068, 25 and 27 genes for 10 and 50 ppb, respectively), mTOR signaling pathway (ame04150, 16 genes, 50 ppb), Hippo signaling pathway—fly (ame04391, 26 and 24 genes for 10 and 50 ppb, respectively), phototransduction—fly (ame04745, 14 genes for both 1 and 50 ppb), and insulin resistance (ame04931, 21 and 22 genes for 10 and 50 ppb, respectively). For downregulated DEGs, only tRNA processing (GO:0008033, 9 genes, 50 ppb) was significantly enriched in GO analysis, and the KEGG pathways were Fanconi anemia pathway (ame03460, 15 genes, 50 ppb), fatty acid degradation (ame00071, 13 genes, 50 ppb), fatty acid elongation (ame00062, 9 genes, 1 ppb), fatty acid metabolism (ame01212, 17 genes, 1ppb), and metabolic pathways (ame01100, 172 genes, 1 ppb). Upregulation of the Hippo signaling pathway, mTOR signaling pathway, and phototransduction was observed in 14-day-old adults. The Hippo signaling pathway and mTOR signaling pathway are involved in the control of cell proliferation, growth and development [81,82], while phototransduction is the process that converts photons into an action potential in photoreceptor cells (https://www.genome.jp/dbget-bin/www_bget?ame04745; accessed on 29 October 2021).

For 9-day-old larvae, the physiological responses of up-regulated DEGs were unclear, but the down-regulation of Hippo signaling pathway may suggest a defect or delay of development [81]. For 0- and 7-day-old adults, most of the DEGs were enriched in terms involved in multiple functions or fundamental pathways, such as transcription and translation. We did not observe terms related to any physiological response, suggesting that the effect on imidacloprid might not reach the threshold of damaging the downstream pathways or resulting in severe physiological effect. For 14-day-old adults, the upregulation of development-related pathways may be related to a maturation process of individuals, while the downregulation of metabolism pathways and fatty acid metabolism may suggest a defect of metabolism or energy storage, especially of fatty acid. Workers suffering a shortage of energy sources or metabolic disorder may be under stress and have shorter life spans [83]. In addition, the DNA repairing process may also be affected, as the Fanconi anemia pathway, the DNA repairing pathway [84], was found to be downregulated. These differentially expressed terms/pathways may be correlated with imidacloprid treatment during the larval stage, as some of them showed related general functions. Nevertheless, they may also be correlated with the developmental queue of honey bees. The transcriptomic profiles suggested the development of precocious foragers in 14-day-old adults after imidacloprid treatment during the larval stage. The switch from nurse bees to foragers may involve many physiological changes in tissues/organs, and foragers are more positively phototactic [79,85]. Thus, at least three pathways were correlated with the development of honey bees. Either hypothesis or both are possible, as the precocious event is also induced by imidacloprid treatment. The affected GO terms, KEGG pathways, and gene lists after different concentrations of imidacloprid treatments are shown in Appendix A (upregulated genes in 9-day-old larvae), 2 (downregulated genes in 9-day-old larvae), 3 (upregulated genes in 0-day-old adults), 4 (downregulated genes in 0-day-old adults), 5 (upregulated genes in 7-day-old adults), 6 (downregulated genes in 7-day-old adults), 7 (upregulated genes in 14-day-old adults), and 8 (downregulated genes in 14-day-old adults).

To understand the significance and network connection of differentially expressed terms/pathways, overrepresentation analysis was performed for GO biological pathways of upregulated and downregulated DEGs for 9-day-old larvae and 0-, 7-, and 14-day-old adults among different concentrations of imidacloprid treatments (Figure 1a,b). For 9-day-old larvae, upregulated DEGs were enriched in the structural constituents of the ribosome and transmembrane transporter activity among different concentrations of imidacloprid treatment, while those of 0-, 7-, and 14-day-old adults were voltage-gated ion channel activity (0-day), structural molecule activity (7-day), structural constituent of ribosome (7-and 14-day), and helicase activity (7-day) (Figure 1a). Downregulated DEGs enriched in 9-day-old larvae were related to guanyl-nucleotide exchange factor activity, protein kinase activity, calcium ion binding, and protein serine/threonine kinase activity, while those of 0-, 7-, and 14-day-old adults were enriched in oxidoreductase activity (0-day), sequence-specific DNA binding (7-day), structural constituent of the cuticle (7-day), nuclease activity (14-day), and isomerase activity (14-day) (Figure 1b). The top 10 enriched GO pathways are shown in Appendix A.

To provide better insights into how those DEGs are involved in significant terms, considering the potential biological complexities in which a gene may belong to multiple annotation categories, to provide information on numeric changes if available, and to better analyze the missing/gaining pathway functions, further analysis was performed. We provided centplots (Figure 2a,b) [86,87] to extract the complex association, which depicts the linkages of genes and biological concepts as a network. Although the quantity of DEGs identified in 0-day-old adults was nearly the same as those identified in 9-day-old larvae and greater than those identified in 7-day-old adults, the gene set size of 0-day-old adults in the upregulated group was significantly smaller in the representative pathways compared to 9d larvae and 7d and 14d adults (Figure 2a). The gene set size corresponding to the enriched pathways and thus the DEGs in 0-day-old adults were enrolled in related, simple, and few functional pathways compared to 9-day-old larvae and 7-day-old adults. This may be correlated with the task performance status of the worker, as 0-day-old workers participate only in simple tasks [88], or the effect of imidacloprid was diminished during the process of metamorphosis. In addition, we observed that the downregulated DEGs of 0-day-old adults lacked oxidoreductase activity (GO:0016491) but recovered in 14-day-old adults. Oxidoreductase activity (GO:0016491) is an important biochemical reaction that catalyzes the transfer of electrons from one molecule to another, suggesting that high electron transfer activity was required for 14-day-old adults. For downregulated DEGs in 7-day-old adults, we also found downregulation of the structural constituent of the cuticle. Exposure to 5 or 200 ppb imidacloprid affected the cuticle proteolysis of honey bees, and the effect was cast-dependent [89]. The downregulation of cuticle constituents is highly likely to be correlated with the affected cuticle proteolysis. Some unknown structural reconstitution or regeneration may occur in 7-day-old adults, and the process would be hampered due to structural defects.

GO enrichment networks organized on the basis of connecting overlapping gene sets were predicted for DEGs identified at different worker ages. For upregulated DEGs of 9-day-old larvae, most of the functional terms were connected, and networks correlated with terms of ion transmembrane transporter activity and ion transmembrane activity. For 0-, 7-, and 14-day-old adults, the network connections for different adult ages were tightly coordinated with ion channel activity (0-day-old), signaling/enzyme receptor (0-day-old), voltage-gated ion channel activity (0-day-old), ribonucleotide binding (7-day-old), ATP-binding/ATPase-coupled transmembrane transporter activity (7-day-old), helicase activity (7-day-old), ion/membrane transporter activity (14-day-old), and gated channel activity (14-day-old) (Figure 3a). There were some insignificant and isolated terms, such as NAD binding (0-day-old), cytoskeletal protein binding (7-day-old), lyase activity (7-day-old), actin-binding (14-day-old), and mRNA binding (14-day-old). For downregulated DEGs, the network connections among all GO terms were not as highly connected as those among upregulated DEGs. For 9-day-old larvae, the GO terms catalytic activity, protein kinase activity, and ribonucleotide binding showed complex connections. For different adult ages, the GO terms of peptidase activity (0-day-old), unfolded protein binding (0-day-old), transferase activity (7-day-old), drug binding (7-day-old), catalytic activity (7-day-old), protein kinase activity (7-day-old), phosphotransferase activity, alcohol group as receptor (7-day-old), catalytic activity (acting on RNA) (14-day-old), and nuclease activity (14-day-old) showed complex connections with other terms (Figure 3b). Appendix A illustrate the GO term networks with directional connections for upregulated and downregulated DEGs, respectively.

After exposure to 500 ppb imidacloprid during the larval stage, developed adults showed a substantial impact on the expression of detoxification, metabolism, neuron development, structural protein, and transporters and receptors [66]. While providing 20 ppb imidacloprid during the larval stage, immune pathway-related genes were significantly downregulated in pupae and newly emerged adults [67]. From the perspective of the developmental queue, the affected genes and functional pathways were diverse among workers of different ages. Nevertheless, they seemed to be coordinated with bee development and task performance and may consequently result in precocious foragers [79].

## 5. The Molecular Effect of Imidacloprid on Other Pollinators

Imidacloprid affects other pollinator bees, in addition to honey bees. Bumble bees seem to be very sensitive to imidacloprid. Ten ppb imidacloprid caused reductions in foraging activity, locomotion, and even foraging rhythmicity in *Bombus terrestris* [90,91]. Exposure to 7.5 ppb imidacloprid for 4 continuous d resulted in a weak influence. Only 1 gene was differentially expressed, 1 gene showed alternative splicing in bumble bee workers, and 8 genes had alternative splicing in queen bumble bees [90]. Bebane et al. profiled the methylome and transcriptome of bumble bee (*B. terrestris*) workers after exposure to realistic field imidacloprid concentrations (10 ppb) for 6 d. A total of 405 genes were differentially expressed (192 upregulated, 213 downregulated), while there was no significant effect on DNA methylation. DEGs were enriched in functional terms related to apoptotic processes (33 genes), energy reserve metabolism (6 genes), immune response (3 genes), negative regulation of cell communication (24 genes), oxidation-reduction process (37 genes), and P450 genes (4 genes), as well as synaptic transmission, xenobiotic metabolism, and resistance to insecticides (28 genes) [91]. A transcriptomic approach of the Asian honey bee *Apis cerana* revealed that consuming a sucrose solution containing LC5 (0.968 ppm) of imidacloprid for 24 h resulted in 709 DEGs among 1-, 8-, and 16-h-post treatments. The upregulation of genes was related to responses to stimuli, localization, transporter activity, and signal transducer activity, while downregulated genes were enriched in terms related to metabolic process, detoxification (cytochrome P450), catalytic activity, and structural molecule activity. Additionally, a KEGG analysis suggested the upregulation of the phenylalanine metabolism pathway, FoxO signaling pathway, mTOR signaling pathway, and tyrosine metabolism pathway. Downregulated pathways after imidacloprid exposure were protein processing in the endoplasmic reticulum, arginine and proline metabolism, and glycine, serine, and threonine metabolism [92]. Most of the studies focused on the effects on eusocial bees; very few reported the effect on solitary bees, especially the molecular effect. As the model species for the study of solitary bees, the red mason bee, *Osmia bicornis,* was found to be very sensitive to imidacloprid [93]. Beadle et al., (2019) sequenced the genome of *O**. bicornis* and confirmed that *O**. bicornis* lacks the superfamily of P450 enzyme but contains a P450 within the CYP9BU superfamily [93]. This enzyme metabolizes thiacloprid but not imidacloprid. *O**. bicornis* is thus thought to be highly sensitive to imidacloprid and can be killed with a relatively low dosage (LD_50_ 0.046 μg/bee). Exposure to 0.0001 μg of imidacloprid resulted in a total of 27 genes that were upregulated, while 52 genes were downregulated. There was no P450 related gene identified after imidacloprid treatment; however, the authors suggest that constitutive rather than changed P450 expression levels were more important for the detoxification process after pesticide exposure [93]. Several affected molecular pathways, such as metabolism, immune response, and detoxification/P450, are found in western honey bees, Asian honey bees, and bumble bees. Although responses vary among different species of insects, a universal effect can still be established on the basis of current understanding.

## 6. Conclusions

In conclusion, the lethal dose/concentration and molecular effects of sublethal dosages of imidacloprid on honey bees and other pollinator bees were integrated and reviewed. Molecular evidence suggests that the expression of the immune response, detoxification, oxidation-reduction, and other development-related genes was ubiquitously affected among different species of target bees. Transcriptomic approaches revealed that even very low dosages/concentrations of imidacloprid could cause global effects, even altering the developmental queue and inducing a precocious forager. Realistic field levels of imidacloprid severely impact the sustainable development and population dynamics of domesticated and wild pollinators. The application of imidacloprid and other neonicotinoid pesticides should be more carefully and rigorously evaluated.

## Figures and Tables

**Figure 1 ijms-22-11835-f001:**
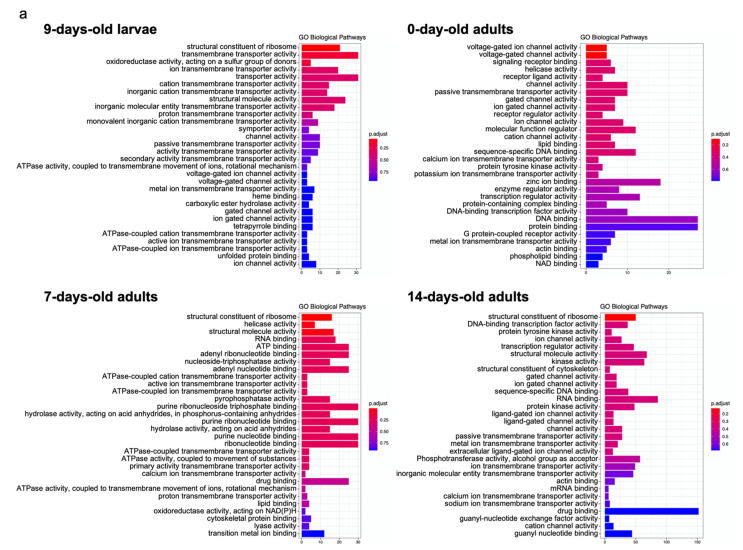
GO biological pathways of (**a**) upregulated and (**b**) downregulated DEGs among different imidacloprid treatments of 9-day-old larvae, 0-day-old adults, 7-day-old adults, and 14-day-old adults. *X*-axis: number of genes; *Y*-axis: GO terms; p.adjust: the adjusted p value of identified GO terms; bar color from red to blue represents the adjusted p value from low to high.

**Figure 2 ijms-22-11835-f002:**
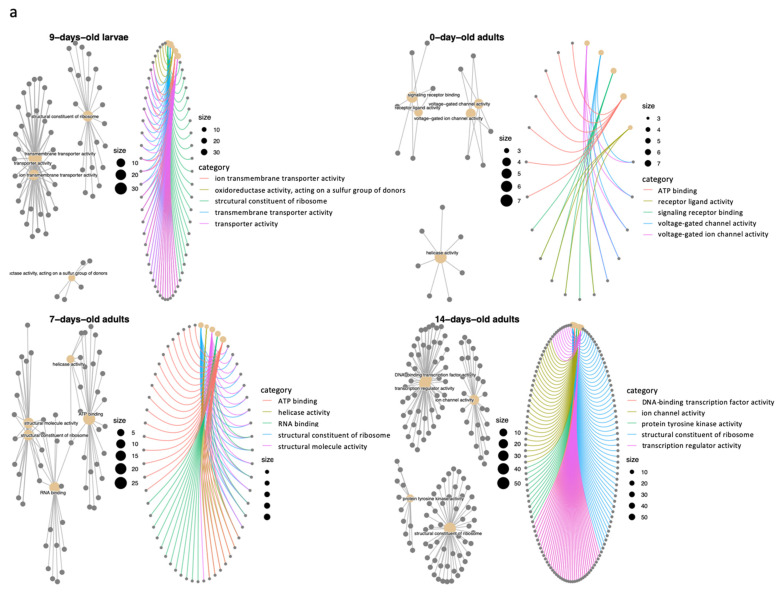
Gene-pathway concept network for overrepresentation testing for GO terms in the pool of (**a**). upregulated and (**b**). downregulated DEGs among different imidacloprid treatments in 9-day-old larvae, 0-day-old adults, 7-day-old adults, and 14-day-old adults. Size: number of DEGs; category: the GO terms category. Gray lines indicate the network connection, and the weight of the line reflects the degree of connection between terms. Colored lines indicate the connection between terms and the involved genes. The concept network plot shows that the genes are involved in significant terms and biological complexness.

**Figure 3 ijms-22-11835-f003:**
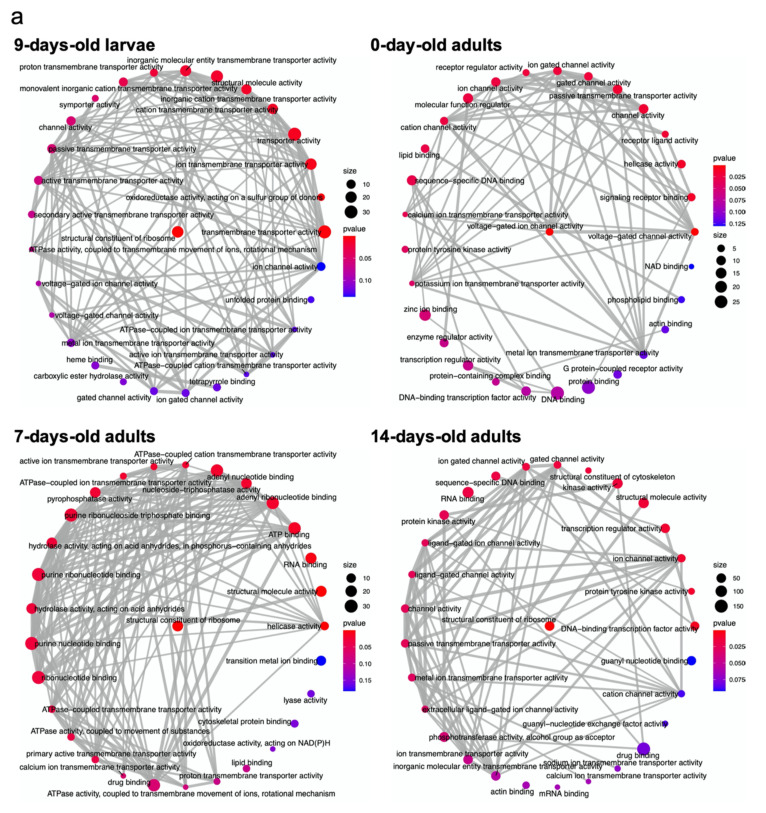
Enrichment maps for overrepresentation testing for GO terms in the pool of (**a**) upregulated and (**b**) downregulated DEGs among different imidacloprid treatments in 9-day-old larvae, 0-day-old adults, 7-day-old adults, and 14-day-old adults. Size: number of DEGs; p value: the p value of identified GO terms. The color from red to blue represents the p value from low to high. Gray lines indicate the network connection, and the weight of the line reflects the degree of connection between terms.

**Table 1 ijms-22-11835-t001:** LD_50_ and LC_50_ results for imidacloprid in honey bees (Apis mellifera).

	Oral LD50a (ng Per Bee)	Contact LD50 (ng Per Bee)	Oral LC50a (ppm)	Test Period	Honey Bee	Tested Areas	Hours	Reference
adults	NA	40	NA	NA	NA	USA	24	[37]
	5.4 (5.2–1.6)	23.8 (22.3–25.3)	NA	NA	*Apis mellifera mellifera*	France	24	[24]
	6.6 (5.1–8.1)	15.1 (11.9–18.3)			*Apis mellifera caucasica*			
	4.8 (4.5–5.1)	24.3 (22.0–26.6)	NA	NA	*Apis mellifera mellifera*		48	
	6.5 (4.7–8.3)	12.8 (9.7–15.9)			*Apis mellifera caucasica*			
	41	NA	NA	Jul-99	NA	Germany I	72	[26]
	20	104 (83.0–130)		Jul-99		The Netherlands I		
	81	61.0 (26.0–90.0)		May-00		Germany II		
	81	50.0 (9.1–71.0)		May-00		United Kingdom I		
	81	42.0 (20.0–59.0)		May-00		Germany III		
	81	42.9 (34.6–53.2)		May-00		Germany IV		
	81	74.9 (61.8–90.9)		Jul-00		Germany V		
	57 ± 28	NA	NA	NA	NA	France	48	[25]
	37 ± 10						72	
	37 ± 10						96	
	3.7 (2.6–5.3)	81 (55.0–119.0)	NA	NA	NA	UK	48	[2]
	> 21.0	230.3				Netherlands		
	40.9	nt				Germany		
	11.6 (7.3-18.3)	242.6 (173.3–353.4)						
	21.2 (15.0-29.6)	59.7 (39.1–92.7)						
	NA	17.9 (9.2–31.5)	NA	Jun to Sep-99	NA	USA	24	[38]
	30.6	NA	NA	NA	NA	France	48	[39]
	25.4 ± 22.8	NA	NA	NA	NA	France	48	[40]
	118.74	NA	NA	NA	NA	Italy	24	[23]
	90.09						48	
	69.68						72	
	NA	29	0.003	NA	NA	Egypt	24	[27]
		26	0.0006				48	
larvae	4170 (2960–5850)	NA	138.84 (98.20–196.30)	NA	NA	USA.	72	[28]
	1400		NA	NA	NA	Taiwan	268	[41]

**Table 2 ijms-22-11835-t002:** (**a**). Differentially expressed genes of the honey bee after imidacloprid treatment during the adult stage. (**b**) Differentially expressed genes of the queen bee after imidacloprid treatment during the adult stage. (**c**) Differentially expressed genes of honey bees after imidacloprid treatment during the larval stage.

(a)													
Imidacloprid	Treatment		Approaches	Treatment Stages		Volume	Affected Functions	Gene id/Name	Gene Regulatory Trend		Treatment Period (d)	Sampling d (d Post First Day of Treatment)	References
	Treatment	Raised Environment							Up	Down			
20 ppb	feeding (syrup)	NA		adults		NA	NA	Lim 3 homeobix (down),	0	1	7	7	[63]
								vanin-like protein 1-like (2) (down)	0	1		11	
20 ppb	topical treatment	caged	qPCR	adults		1 μL	P450 superfamily genes	CYP6AS14	0	1	1	1	[60]
							immunity	thioredoxin peroxidase, Amel/LRR	0	1		1	
								eater, Amel/LRR	0	2		7	
							development and antioxidant enzyme-coding gene	Catalase (down), SOD (up)	1	1		1	
5 ppb	feeding (pollen patty and sugar solution)	field	qPCR	adults		NA	immunity	Apisimin (down), Defensin 2 (down), Vitellogenin (up)	1	2	10	10	[61]
		caged						Abaecin (down), Apisimin (down), Defensin 1 (down), Defensin 2 (down)	0	4			
		field						Abaecin (down), Apisimin (up)	1	1	20	20	
		caged						Abaecin (down), Defensin 1 (up), Vitellogenin (down)	1	2			
200 ppb		field						Defensin 2 (up), Vitellogenin (up)	2	0	10	10	
		caged						Abaecin (down), Apisimin (down), Defensin 1 (down), Defensin 2 (down), Vitellogenin (down)	0	5			
		field						Abaecin (up), Defensin 2 (up), Vitellogenin (up)	3		20	20	
		caged						Abaecin (down), Apisimin (down)	0	2			
5 ppb		field					detoxification genes	AChE-1 (down), CYP6AS3 (down), CYP9Q3 (down)	0	3	10	10	
		caged						AChE-1 (down), AChE-2 (down), CYP6AS4 (down), CYP6AS10 (down), CYP9Q1 (down), CYP9Q2 (down), CYP9Q3 (down), CYT p450 (down)	0	8			
		field						AChE-2 (down), CYP6AS3 (down)	0	2	20	20	
		caged						AChE-1 (down), CYP6AS3 (down), CYP6AS4 (down), CYP9Q1 (up), CYP9Q2 (down), CYP9Q3 (up), CYT p450 (up)	3	4			
200 ppb		field						AChE-2 (down), CYP6AS10 (down)	0	2	10	10	
		caged						AChE-1 (down), AChE-2 (down), CYP6AS4 (up), CYP6AS10 (down), CYP9Q1 (down), CYP9Q2 (down), CYP9Q3 (down)	1	6			
		field						AChE-1 (up), AChE-2 (up), CYP6AS3 (up), CYP9Q1 (up), CYP9Q2 (up), CYP9Q3 (up), CYT p450 (up)	7	0	20	20	
		caged						AChE-2 (up), CYP6AS3 (down), CYP6AS10 (down), CYP9Q1 (up), CYP9Q2 (down), CYT p450 (down)	2	4			
5 ppb	sugar syrup	caged	aqPCR	adults		NA	detoxification enzyme	GstD1	1	0	7 weeks	7 weeks	[59]
20 ppb													
100 ppb													
10 ppb	feeding, (sucrose solution, 50% wt/wt)	caged	NGS	adults		NA	total DEGs 509	NA	160	349	8	8	[64]
							ribosomal protein		28	0			
							phototransduction		NA				
							visual perception						
							photoreceptor						
							actin binding						
							actin cytoskeleton						
							muscle attachment						
							somatic muscle development						
5 ppb	feeding (food patty)	caged	qPCR	adults		NA	antioxidant genes	Cat (down), MsrA (down), TrxR1 (down)	0	3	10	10	[62]
10 ppb							antioxidant genes	Cat (up), TrxR1 (up), SelK (up), MsrB (up), Sod2 (down)	4	1			
20 ppb	feeding (sucrose solution, 30% wt/wt)	caged	NGS	adults		NA	total DEGs	131	1	130	11	11	[53]
							chemosensory-related genes	GB46225, GB46227, GB46230, GB53372, GB16006, GB50003	0	6			
							immune and detoxification response	abaecin (GB18323), apisimin (GB53576), defensin 1 (GB41428), glucose dehydrogenase (GB43007), glucose dehydrogenase-like (GB51446), leucine-rich repeat-containing protein 26-like (GB44192), phenoloxidase subunit A3 (GB43738), serine protease easter (GB45700) and tyrosine aminotransferase (GB45969)	0	8			
							insecticide resistance-related gene	cuticular protein 14 (GB46297), cytochrome b561 (GB40148), cytochrome P450 6a2 (GB49876), cytochrome P450 9e2 (GB43713), esterase A2 (GB43571), cytochrome P450 6a17 (GB49885) and UDP-glucuronosyltransferase 2C1 (GB52179)	0	7			
							oxidation-reduction	GB52785, GB44549, GB43007, GB55515, GB49876, GB50655, GB43713, GB51446, GB50178, GB49885	0	10			
							iron ion	GB55515, GB49876, GB50655, GB43713, GB49885	0	5			
							oxidoreductase activity	GB52785, GB44549, GB43007, GB55515, GB49876, GB50655, GB43713, GB51446, GB50178, GB49885	0	10			
							behavioral response	SLC18A2 (GB50003), melanopsin (GB41643), aquaporin 4 (GB41240), PRKACB (GB48362)	0	4			
							phototransduction	GB41297, GB51068	0	2			
0.3 ng/bee	sucrose solution	caged	NGS	adults		100 μL/bee in average per 24 h	metabolic pathways	DN74754_c11_g2 (down)	7	19	2	2	[65]
							starch and sucrose metabolism	DN74754_c11_g2 (down)					
							purine metabolism	DN75806_c0_g2 (down)					
3 ng/bee							metabolic pathways	DN75371_c1_g1 (down), DN178528_c0_g1 (up), DN74754_c11_g2 (down)	36	77			
							pentose phosphate pathway	DN75371_c1_g1 (down)					
							purine metabolism	DN75806_c0_g2 (down)					
							glycine, serine, and threoninemetabolism	DN178528_c0_g1 (up)					
							porphyrin metabolism	DN178528_c0_g1 (up)					
							starch and sucrose metabolism	DN74754_c11_g2 (down)					
(b)													
imidacloprid	treatment		Approaches	treatment stages		volume	affected functions	Gene ID/name	DEGs trend		treatment period (days)	sampling days (days post first day of treatment)	references
	treatment	raised environment							up	down			
20 ppb	topical treatment	caged	qPCR	queen, adults		2 μL	P450 superfamily genes	CYP306A1 (down)	0	1	1	1	[60]
								CYP4G11 (up), CYP6AS14 (down)	1	1		7	
							immunity	thioredoxin peroxidase (up), apidaecin (down), eater (down), Amel/LRR (down), VgMC (down)	1	4		1	
								thioredoxin peroxidase (up), Amel/LRR (down), VgMC (down)	1	2		7	
							development and antioxidant enzyme-coding gene	SOD (down), hexamerin 70b (down)	0	2		1	
								hexamerin 70b (down)	0	1		7	
(c)													
imidacloprid	treatment		Approaches	treatment stages	collected stages	volume	affected functions	Gene ID/name	DEGs trend		treatment period (d)	sampling d (d post first day of treatment)	references
	treatment	raised environment							up	down			
500 ppb	feeding (water)	field	NGS	larvae	adults	1 μL	total DEGs	578			4	21	[66]
							detoxification	GB48993 (up), GB49876 (up), GB43693 (up), GB44513 (up), GB55257 (up), GB49877 (up), GB49614 (up), GB48905 (down), GB49875 (down), GB40287 (down), GB43728 (down), GB49887 (down), GB52023 (down), GB43716 (down), GB43727 (down), GB51356 (down), GB47279 (down), GB49885 (down), GB55669 (down), GB49626 (down), GB46814 (down), GB43713 (down), GB49886 (down), GB43715 (down),	7	17			
							immunity	GB51223 (down), GB47318 (down), GB47546 (down), GB40164 (down), GB53576 (down)	0	5			
							mitochondria	GB44116 (up), GB42580 (up), GB50970 (up), GB42141 (up), GB49306 (up), GB46083 (up), GB53201 (up), GB49942 (down), GB47970 (down), GB42550 (down), GB51583 (down)	7	4			
							metabolism	GB42460 (up), GB55040 (up), GB54404 (up), GB49562 (down), GB51236 (down), GB54302 (down), GB49336 (down), GB48172 (down), GB53312 (down), GB51247 (down), GB51580 (down), GB51815 (down), GB45596 (down), GB53525 (down), GB54401 (down), GB48850 (down), GB49380 (down), GB51814 (down), GB53579 (down), GB54396 (down), GB43006 (down), GB44548 (down), GB43247 (down), GB53872 (down)	3	21			
							neuron development	GB50170 (up), GB50061 (up), GB42798 (up), GB41270 (up), GB49750 (up), GB47563 (up), GB47918 (up), GB55389 (up), GB41630 (up), GB47565 (up), GB51612 (up), GB52630 (up), GB46091 (up), GB49726 (up), GB41126 (down), GB52454 (down), GB41856 (down), GB43778 (down), GB40356 (down), GB43504 (down), GB49109 (down), GB43788 (down), GB49708 (down)	4	19			
							sensory processing	GB50936 (up), GB55547 (up), GB45850 (up), GB52326 (up), GB41643 (down), GB46229 (down), GB40616 (down), GB44550 (down), GB46225 (down),GB46230 (down), GB53367 (down), GB51369 (down), GB46224 (down), GB46228 (down), GB53368 (down), GB53372 (down), GB54970 (down),GB51189 (down)	4	14			
							signaling pathway	GB49937 (up), GB41862 (down), GB50705 (down), GB47566 (down), GB47961 (down), GB48815 (down), GB41220 (down), GB51125 (down)	2	6			
							structural protein	GB47903 (up), GB42581 (up), GB50453 (up), GB44002 (up), GB49021 (up), GB50438 (up), GB53565 (up), GB53119 (up), GB50236 (up), GB45073 (up), GB40566 (up), GB49845 (up), GB41227 (up), GB41203 (up), GB45174 (up), GB41311 (up), GB47902 (up), GB41310 (up), GB45968 (up), GB41308 (up), GB44214 (up), GB52194 (up), GB45211 (up), GB41946 (up), GB45943 (up), GB52014 (up), GB52161 (up), GB43173 (up), GB40253 (up), GB44074 (up), GB42571 (up), GB49219 (up), GB43377 (down), GB51391 (down), GB40304 (down), GB41757 (down)	32	4			
							transcription factors	GB50933 (up), GB52658 (up), GB50795 (down), GB42049 (down), GB52761 (down)	2	3			
							transporters and receptors	GB47513 (up), GB42142 (up), GB53053 (up), GB54918 (up), GB42802 (up), GB43870 (up), GB42865 (up), GB40973 (up), GB47391 (up), GB49727 (up), GB40867 (down), GB43672 (down), GB50423 (down), GB54881 (down), GB47942 (down), GB44824 (down), GB48790 (down), GB49801 (down), GB44987 (down), GB54942 (down), GB49396 (down), GB46030 (down), GB41240 (down), GB50098 (down), GB48330 (down), GB51487 (down), GB47278 (down), GB49473 (down), GB40818 (down), GB42942 (down), GB47931 (down), GB45986 (down), GB54467 (down), GB43963 (down), GB41182 (down), GB50890 (down), GB55098 (down), GB55239 (down), GB51504 (down), GB53124 (down), GB50457 (down), GB52097 (down), GB41815 (down), GB55503 (down), GB46597 (down), GB45159 (down)	10	36			
20 ppb	feeding (larvae diet)	caged	qPCR	larvae	white eye pupae	10 μL	Toll pathway	Cactus (down), Dorsan (down)	0	2	4	9	[67]
							IMD pathway	PGRP LC710 (down)	0	1			
							JNK pathway	Kayak (down)	0	1			
							Antimicrobial peptides	Abeacin (down)	0	1			
							Melanization	PPO (up)	1	0			
					brown eye pupae		Toll pathway	PGRP SC 4300 (down), Spaetzle (down), Cactus (down), Dorsan (down)	0	4		12	
							JAK/STAT	Domless (down)	0	1			
							IMD pathway	PGRP LC710 (down), Relush (down)	0	2			
							JNK pathway	Basket (down), Kayak (down)	0	2			
							Antimicrobial peptides	Apidaecin (down), Hymenoptaecin (down), Defensin 1 (down)	0	3			
							Melanization	PPO (down)	0	1			
					newly emerged adults		Toll pathway	Cactuc (up)	1	0		16	
							IMD pathway	Relish (up)	1	0			
							JNK pathway	Kayak (up)	1	0			
							Antimicrobial peptides	Hymenoptaecin (up), Lysozyme 2 (up)	2	0			
							Melanization	PPO (up)	1	0			

## Data Availability

Honey bee RNA-seq raw reads are available at https://www.ncbi.nlm.nih.gov/sra/PRJNA521949 accessed on 27 October 2021.

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
