# Peer review of "Chronic Effects of Imidacloprid on Honey Bee Worker Development—Molecular Pathway Perspectives"

_ijms, 2021, doi:10.3390/ijms222111835_

Round 1

Reviewer 1 Report

The publication provides a broad description of the effect of imidacloprid on the honey bee.

Several points need to be corrected and / or detailed:

  • Page 2 - The authors provide LD50 for imidacloprid in various countries. It would be worth justifying where the differences in these values ​​in individual countries come from.
  • What the authors mean by "on the ecological sustainability of honey bees". It is worth explaining this.
  • The authors in the tables and several times in the text use the abbreviations DEG. Explain this abbreviation the first time you use it.
  • In Figure 1 (a and b) the markings / letters of the descriptions (left) are too small for each developmental / age stage of the bees. Descriptions need to be enlarged to make them legible.
  • Figure 2 - The categories description is too small. Enlarge these descriptions.
  • In Chapter 4, the authors list and describe the genes whose expression changes after imidaclopride. In some places, they add information about how this affects the functioning of proteins / enzymes (e.g. proteases and oxidoreductases). To improve citation of this review, please complete the information on the effects of imidacloprid on other proteins / enzymes / mechanisms in terms of gene expression.
  • Chapter 5 should be more detailed. It would be useful to add a description of the effect of imidacloprid on other pollinators, such as Osmia bicornis; and not only bumblebees and A. cerana. The description does not have to be long, but it is worth drawing readers' attention to the seriousness of the problem and signaling that insecticides are a serious problem for pollinators and the ecosystem.
  • There are different font sizes in References.

Author Response

reviewer 1

The publication provides a broad description of the effect of imidacloprid on the honey bee.

Several points need to be corrected and/or detailed:

  • Page 2 - The authors provide LD50 for imidacloprid in various countries. It would be worth justifying where the differences in these values in individual countries come from.

Author’s response: We have added some studies related to different insecticide sensitivity of honey bees among different regions.

  • What the authors mean by "on the ecological sustainability of honey bees". It is worth explaining this.

Author’s response: The ecological sustainability of honey bee and other pollinators after exposure to sublethal dosage of imidacloprid is an important and critical issue and have been well-reviewed. We added the reason of impact and added more citations to address this phenomenon.

  • The authors in the tables and several times in the text use the abbreviations DEG. Explain this abbreviation the first time you use it.

Author’s response: Thank you for your suggestion. We have added the full name of DEG

  • In Figure 1 (a and b) the markings / letters of the descriptions (left) are too small for each developmental / age stage of the bees. Descriptions need to be enlarged to make them legible.

Author’s response: Thank you for your suggestion. We have enlarged the description.

  • Figure 2 - The categories description is too small. Enlarge these descriptions.

Author’s response: Thank you for your suggestion. We have enlarged the description.

  • In Chapter 4, the authors list and describe the genes whose expression changes after imidacloprid. In some places, they add information about how this affects the functioning of proteins/enzymes (e.g. proteases and oxidoreductases). To improve citation of this review, please complete the information on the effects of imidacloprid on other proteins / enzymes / mechanisms in terms of gene expression.

Author’s response: Thank you for your suggestion. We have put more descriptions for potential physiological effects on the honey bee.

  • Chapter 5 should be more detailed. It would be useful to add a description of the effect of imidacloprid on other pollinators, such as Osmia bicornis; and not only bumblebees and A. cerana. The description does not have to be long, but it is worth drawing readers' attention to the seriousness of the problem and signaling that insecticides are a serious problem for pollinators and the ecosystem.

Author’s response: Thank you for your suggestion. We have put more descriptions for effects on Osmia bicornis.

  • There are different font sizes in References.

Author’s response: Thank you for your suggestion. We have corrected it.

Reviewer 2 Report

This interesting paper deals with chronic effects of imidacloprid on honey bee worker development providing a molecular pathway perspective.

This review collated current knowledges on lethal dose/concentration and molecular effects of sublethal dosages of imidacloprid on honey bees and other pollinator bees.

Based on molecular evidences, the expression of the immune response, detoxification, oxidation-reduction, and other developmental-related genes appear ubiquitously affected among different species of target bees.

Transcriptomic investigations revealed that even very low dosages/concentrations of imidacloprid could cause global effects, even altering the developmental queue and inducing a precocious forager.

Furthermore, real field levels of imidacloprid severely impact the sustainable development and population dynamics of managed and wild pollinators.

The review is well structured and organised. Adequately readable despite the complexity of the topic.

Minor changes are indicated in the text.

Author Response

reviewer2

This interesting paper deals with chronic effects of imidacloprid on honey bee worker development providing a molecular pathway perspective.

This review collated current knowledges on lethal dose/concentration and molecular effects of sublethal dosages of imidacloprid on honey bees and other pollinator bees.

Based on molecular evidences, the expression of the immune response, detoxification, oxidation-reduction, and other developmental-related genes appear ubiquitously affected among different species of target bees.

Transcriptomic investigations revealed that even very low dosages/concentrations of imidacloprid could cause global effects, even altering the developmental queue and inducing a precocious forager.

Furthermore, real field levels of imidacloprid severely impact the sustainable development and population dynamics of managed and wild pollinators.

The review is well structured and organized. Adequately readable despite the complexity of the topic.

Minor changes are indicated in the text.

Author’s response: Thank you very much for your comments! We appreciate your help. However, we don’t receive the file of the text with reviewer’s comments.